# Lime-based supplement reduces calcium oxalate stone recurrence: A multicenter randomized controlled trial

Thasinas Dissayabutra[1,2]*, Weerapat Anegkamol[1], Supoj Ratchanon[3],
Wattanachai Ungjaroenwathana[4], Tasanee Klinhom[4], Thosaphol Sasivongsbhakdi[5],
Pisitpol Siriwattana[6], Anuthep Burami[7], Ukrit Wayakkanont[8], Pisit Prapunwattana[2],
Piyaratana Tosukhowong[2]

1 Metabolic Disease in Gastrointestinal and Urinary System Research Unit, Department of Biochemistry, Faculty of Medicine, Chulalongkorn University, Bangkok, Thailand, 2 Department of Biochemistry, Faculty of Medicine, Chulalongkorn University, Bangkok, Thailand, 3 Division of Urology, Department of Surgery, Faculty of Medicine, Chulalongkorn University, Bangkok, Thailand, 4 Division of Urology, Department of Surgery, Sunpasitthiprasong Hospital, Ubon Ratchathani, Thailand, 5 Division of Urology, Department of Surgery, Nopparat Rajathanee Hospital, Bangkok, Thailand, 6 Division of Urology, Department of Surgery, Phayao Hospital, Phayao, Thailand, 7 Division of Urology, Department of Surgery, Burapha University Hospital, Chonburi, Thailand, 8 Division of Urology, Department of Surgery, Sakon Nakhon Hospital, Sakon Nakhon, Thailand

* thasinas@chula.md

## Abstract

### Background

Recurrent urolithiasis is a major clinical challenge, with more than 50% of patients experiencing recurrence within 5 years. While potassium citrate effectively reduces recurrence, poor adherence due to cost and gastrointestinal side effects limits its long-term use. Citrus-based interventions, such as lime juice, have shown potential in enhancing urinary citrate and alkalinity but require further validation. This study evaluated the efficacy of a lime-based phytochemical-rich regimen (LPR) in preventing stone recurrence and reducing urinary inflammation in post-operative urolithiasis patients.

### Objective

This multicenter, double-blind, randomized controlled trial aimed to evaluate the efficacy and safety of a novel lime-based preparation called LPR in preventing kidney stone recurrence over 24 months.

### Methods

In a double-blind, randomized, placebo-controlled, multicenter trial, 173 patients with calcium oxalate urolithiasis who had undergone successful stone removal were enrolled from six hospitals in Thailand. Participants were randomized to receive

**Data availability statement:** https://www.mediafire.com/file/su883wl3xfpqole/Raw+Data+LPR3.xlsx/file.

**Funding:** National Research Council of Thailand, grant number 2556-29 and Thailand Science Research and Innovation Fund, Chulalongkorn University, grant number HEAF67300067.

**Competing interests:** The authors have declared that no competing interests exist.

either LPR or placebo for 24 months. The primary outcome was the incidence of stone recurrence confirmed by computerized topography (CT). Secondary outcomes included changes in urinary protein excretion and urinary interleukin-8 (IL-8) level, a pro-inflammatory cytokine implicated in renal inflammation and stone formation. Kaplan–Meier survival analysis and multivariate Cox regression were used to assess recurrence risk.

### Results

Of 173 enrolled participants, 151 completed the study. The recurrence rate at 2 years was significantly lower in the LPR group (14%) compared to placebo (45%) ($p < 0.001$). Kaplan–Meier analysis demonstrated a hazard ratio (HR) of 0.24 (95% CI: 0.13–0.44; log-rank $p < 0.0001$) favoring LPR. Among completers, LPR significantly reduced urinary IL-8 level ($p = 0.017$) and 24-hour urinary protein excretion ($p = 0.032$) compared to baseline and placebo. No serious adverse events were reported, and adherence was high in both groups.

### Conclusion

LPR, a lime-based supplement rich in citrate and flavonoids, significantly reduced the 2-year recurrence rate of calcium oxalate stones by approximately 76%. This effect may be mediated by increased urinary citrate excretion, alkalinization, and attenuation of renal inflammation, as evidenced by reduced urinary IL-8 and proteinuria. LPR was well tolerated, with minimal adverse effects, and may serve as a safe, cost-effective adjunct for secondary prevention in patients intolerant to conventional alkali therapy.

### Introduction

Urolithiasis is a highly prevalent condition with increasing global incidence, affecting approximately 4-20% of the population worldwide [1]. The recurrence rate of calcium oxalate (CaOx) stones remains alarmingly high, approaching 82.4% for calcium oxalate monohydrate stone [2,3]. This highlights the urgent need for effective, accessible strategies to prevent stone recurrence.

Multiple urinary metabolic abnormalities, such as hypercalciuria, hyperoxaluria, hypocitraturia, and hypomagnesuria, have been associated with increased risk of stone formation [4–6]. Among these, hypocitraturia plays a key role in lithogenesis by reducing the inhibition of calcium crystal aggregation and growth. Accordingly, potassium citrate therapy is the standard pharmacological intervention to prevent recurrence [7,8]. However, its long-term use is limited by gastrointestinal adverse effects and high cost, particularly in low-resource setting [9,10]. Therefore, an alternative regimen with fewer or no adverse effects is desirable.

Natural dietary sources of citrate such as green tea, raspberry, pomegranate, lemon and lime, have been explored as alternative therapies [11]. Lime (*Citrus*

*aurantifolia*), widely consumed in Southeast Asia, contains high concentrations of citrate and bioactive phytochemicals with potential anti-inflammatory properties [12]. Previous small-scale studies suggest that lime supplementation can increase urinary citrate excretion and urine alkalinity, but evidence for its long-term clinical efficacy in preventing stone recurrence remains limited [13]. Recent advances have also highlighted the role of renal inflammation in stone pathogenesis and recurrence [14,15]. Elevated urinary biomarkers such as proteinuria and interleukin-8 (IL-8) indicate tubular injury and crystal-induced inflammation in urolithiasis, suggesting that anti-inflammatory interventions may offer additional therapeutic benefit [16,17].

We developed a standardized lime-based phytochemical-rich regimen (LPR) (about 55 mEq of citrate content), combining freeze-dried lime powder and lime beverage, aiming to enhance urinary citrate and mitigate inflammation [18,19]. This multicenter, double-blind, randomized controlled trial evaluated the efficacy and safety of LPR in reducing radiographic stone recurrence and improving urinary inflammatory markers in patients with a history of CaOx urolithiasis. The study also explored the utility of IL-8 and urinary protein as non-invasive biomarkers of treatment response.

## Methods

### Study design and enrolment of participants

This multicentric, randomized, double-blind, placebo-controlled trial was conducted at six centers around the country: Sunpasitthiprasong and Sakon Nakhon Hospitals of the northeast region, where the prevalence of urolithiasis is highest, Phayao Hospital of the north region, Nopparat Rajathanee and King Chulalongkorn Memorial Hospital of the central area, and Burapha University Hospital of the east region. Participants were recruited according to the following inclusion criteria: age 18–75 years, presence of a solitary stone of any location in the size in any kidney (except ureters and lower urinary stones), radiological confirmation of the stone (ultrasound, computed tomography, intravenous pyelogram or kidney-ureter-bladder x-ray), and had residual stones ≤4 mm at 1-month post-surgery evaluated by KUB x-ray or ultrasound. Participants with chronic kidney disease, chronic liver disease, a history of coronary artery disease, or those taking any medication that alters urinary metabolic profiles were excluded. A priori sample size and power calculations were conducted during the study planning stage. The calculation assumed a two-year stone recurrence rate of 40% in the placebo group and 15% in the LPR group, corresponding to a relative risk reduction of 0.25; 95%CI 0.14–0.44 [20]. With a two-sided α of 0.05 and 80% power, the required sample size was 145 participants (72 per group). To account for an anticipated dropout rate of 10%, the target enrollment was increased to 160 participants. Full details of the power calculation are provided in the Supplementary.

A total of 173 participants were enrolled in the study during 1 June 2017–30 May 2020, the follow-up during 1 June 2017–30 June 2022 and had provided written consent. Subsequently, they were randomized 1:1 using the research randomizer (https://www.randomizer.org/).

### LPR manufacturing, quality control and supplementation

The lime powder regimen (LPR) was innovated from lime juice and peel. LPR was manufactured by Oui Heng International Healthcare Company Limited, and its composition was analyzed for quality control by the Industrial Metrology and Testing Service Centre (MTC), Thailand Institute of Scientific and Technological Research (TISTR). Each sachet of LPR contained 7.75 g of supplement, providing a citrate dose comparable to standard alkali therapy but with a reduced potassium content. In addition, each sachet delivered 153 mg of flavonoids, including hesperidin, diosmin, and eriocitrin. Relative to conventional potassium citrate prescriptions (30–60 mEq/day), the citrate content in a single sachet falls at the upper end of the recommended therapeutic range, whereas the potassium load remains comparatively lower.

To ensure product safety, heavy metal and microorganism levels were also assessed. The determination of heavy metals (mercury, lead and arsenic) and microbial analysis (including *Staphylococcus aureus*, *Clostridium* spp., *Salmonella* spp.,

*Bacillus cereus* and *Escherichia coli*) were carried out. The product was tested and certified by the Thai National Science and Technology Development Agency (NSTDA), which confirmed that heavy metals were below detection limits, which were within the acceptable limits for Daily Consumption of heavy metals – United States Pharmacopoeia (ALDC-HM USP) reference and no harmful microorganisms were present, ensuring the product's safety (Table S1 in S2 File).

The products were blinded by the manufacturer using sealed containers. Participants received 7.75 g of LPR, or placebo *(maltodextrin which is matched in appearance, and texture)* dissolved in 200–300 ml of water, consumed in 10 minutes every evening between 6 and 9 pm. Follow-up visits occurred at 0, 6, 12, 18 and 24 months to obtain blood and urine samples, radiographs, and clinical data. All participants attended scheduled hospital visits every six months for imaging assessments (ultrasound or CT scan). During these visits, urine and blood samples were collected, ensuring complete data captured for both primary and secondary outcomes.

### Detection of kidney stones and biochemical indicators

The stone composition was analyzed by Fourier-transform infrared (FTIR) spectroscopy. Stone recurrence in patients with urolithiasis was monitored by plain KUB radiography at 6 and 18 months, and computed tomography (CT) at 12- and 24-month follow-up. The recurrence was diagnosed when the preexisting stone grows larger than 4 mm, or new stone was detected by radiography.

Blood samples were collected at 0-, 12- and 24-month to quantify serum levels of alanine aminotransferase (ALT), blood urea nitrogen (BUN), serum creatinine (Cr) and estimated glomerular filtration rate (eGFR) using the automated Alinity ci system at the Department of Laboratory Medicine, Faculty of Medicine, Chulalongkorn University. The 24-hour urine samples were collected at the 0- and 24-month of the study to measure urinary parameters using the automated Alinity ci system and the pro-inflammatory cytokine IL-8 level was assessed with an ELISA kit (Human IL-8/CXCL8 ELISA Kit, ACROBiosystems, Delaware, USA) and compared to the baseline levels. Harms were defined as any undesirable medical occurrences experienced by participants during the study period. Adverse events were systematically assessed at each visit using a structured questionnaire and open-ended questioning. Laboratory results, vital signs, and clinical symptoms were monitored. All events were recorded, categorized as related or unrelated to the intervention, and reviewed by an independent safety committee. Participants who developed urolithiasis with a stone size greater than 4 mm were withdrawn from the study and referred to a urologist for appropriate management.

### Statistical analysis

All randomized participants were included in the analysis. For the primary outcome, time to stone recurrence was evaluated using a Cox proportional hazards model to estimate hazard ratios (HRs) with 95% confidence intervals (CIs). Kaplan–Meier curves were constructed to illustrate stone-free survival.

For continuous variables, data distribution was first assessed using the Shapiro–Wilk test. Parametric data are presented as means ± standard deviations, and non-parametric data as medians with interquartile ranges. Between-group comparisons were performed using independent-sample t-tests or Mann–Whitney U tests, as appropriate. Within-group comparisons were conducted using paired t-tests or Wilcoxon signed-rank tests.

All statistical tests were two-sided, and a p-value < 0.05 was considered statistically significant. Data visualization was performed using bar graphs with error bars representing standard deviations or interquartile ranges. Analyses were conducted using SPSS software, version 21.0 (SPSS Inc., Chicago, IL, USA) and GraphPad Prism version 10.1.1 for macOS (GraphPad Software, Boston, MA, USA).

### Ethical considerations

This study was carried out in accordance with the guidelines detailed in the Declaration of Helsinki and was approved by the Ethical Committee for Research in Human Subjects, Department of Thai Traditional and Alternative Medicine, Ministry

of Public Health, Thailand, number 02-2557. Additionally, this trial has been registered in the National Institute of Health (Thai Clinical Trial Registry: https://www.thaiclinicaltrials.org/` ID: TCTR20250530001, date of registration 30 May 2025). The authors confirm that all ongoing and related trials for this drug are registered.

## Results

### Epidemiology of participants and follow-up

There were 173 participants from six hospitals across four regions of Thailand. Ninety-eight participants were randomly assigned to receive the lime powder regimen (LPR), while seventy-five participants received a placebo. There were no differences in sex, age, residential area, stone type, and stone number between each group (Table 1), and the comparison between complete participants and drop-out in Table S2 in S2 File.

Fig 1 shows the timeline of the study. A total of 22 participants had dropped out (12.2% of the LPR group, and 13.3% of the placebo group), mostly due to migration to seek employment. In total, 151 subjects completed the study.

### Stone recurrence rate

Recurrence occurred in the participants as follows; 2 participants in the LPR group and 4 in the placebo group at a 6-month period; 6 in the LPR group and 9 in placebo group at a 12-month period; 4 in the LPR group and 9 in placebo group at 18-month period; and 1 in the LPR group and 11 in placebo group at the end of the study (Fig 1). Overall, a total of 13 participants in the LPR group (15.1%) and 33 in placebo group (50.8%) were diagnosed with stone recurrence with 24 months. The logarithmic ranking test (Mantel-Cox) showed a Chi-square value of 20.85, with a p-value < 0.0001, indicating a significant difference between the two groups. The hazard ratio for recurrence in the LPR group compared to the placebo group was 0.24 (95% CI: 0.13–0.44; p < 0.0001), relative risk reduction was 76% and absolute risk reduction was 35.7%, demonstrating that the risk of recurrence was significantly lower in the LPR group (Fig 2)

### Liver and kidney biomarkers

No significant changes in ALT, BUN, creatinine and eGFR were observed during 24-month follow-up between each group, imply that no biochemical markers indicating hepatotoxicity or renotoxicity were detected (Fig 3A–3D).

**Table 1. Baseline demographic and clinical characteristics of participants.**

|  | LPR | Placebo | p-value |
|---|---|---|---|
| **Participant (persons)** | 98 | 75 |  |
| **Gender: Male** | 41.90% | 38.50% | 0.799 |
| **Age (year-old)** | 50.20±7.16 | 50.55±9.02 | 0.508 |
| **Residential area** |  |  | 0.854 |
| Northeast region | 70.40% | 58.70% |  |
| North region | 10.20% | 10.70% |  |
| Central region | 10.20% | 13.30% |  |
| East region | 10.20% | 18.70% |  |
| **Type of stone** |  |  | 0.364 |
| Calcium oxalate | 89.90% | 84.00% |  |
| Mixed calcium stone | 10.20% | 16.00% |  |
| **Number of stones present at diagnosis** |  |  | 0.797 |
| 1 stone | 75.50% | 77.30% |  |
| > 1 stone | 24.50% | 22.70% |  |

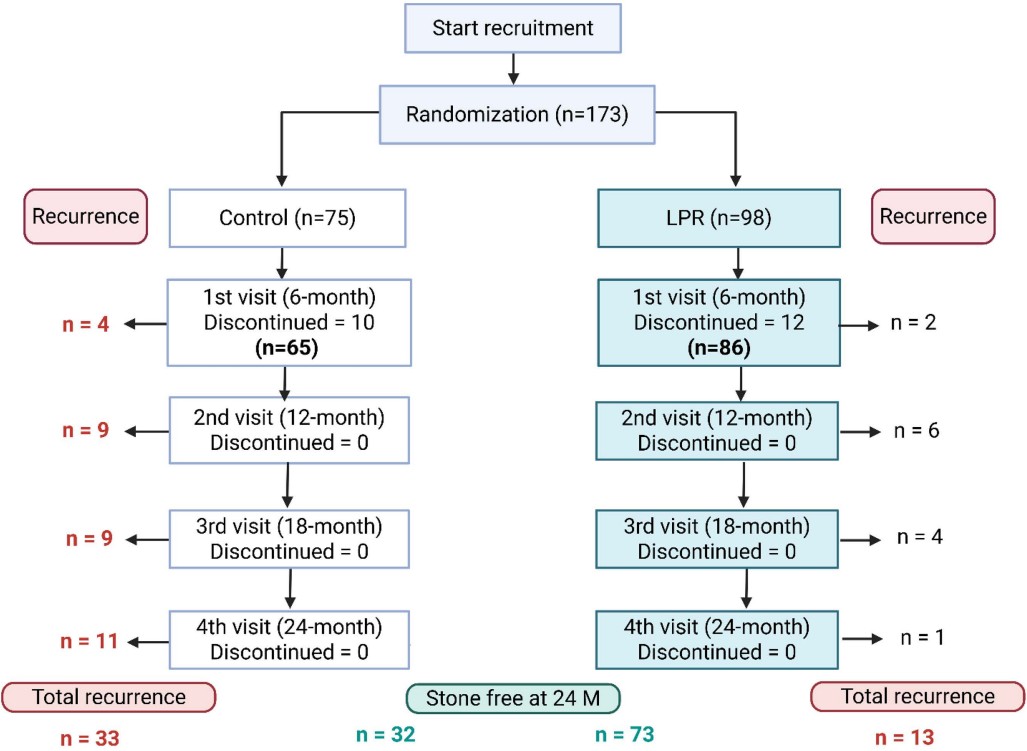

**Fig 1. Timeline and participant flow of the 24-month clinical trial.** The figure illustrates the 24-month clinical trial timeline. The trial began with 173 participants, randomly assigned to the LPR group (98 participants) and the placebo group (75 participants). During six months of follow-up, 22 participants had dropped out and 151 participants remained. At the end of the study, the remaining 105 participants (32 from placebo group and 73 from LPR group) had no stone recurrence detected by CT scan. Figure was created by Biorender.comNo significant adverse effects or gastrointestinal discomfort were observed. However, transient tooth hypersensitivity was reported and resolved with tooth brushing.

## 24-hour urine protein excretion and IL-8 levels

Within-group analysis demonstrated that 24-hour urinary protein excretion and IL-8 levels were markedly reduced in participants receiving LPR compared with baseline (Fig 4). In contrast, no significant changes were observed in the placebo group. Between-group analysis showed no differences in urinary protein or IL-8 levels at baseline; however, at 24 months, both urinary protein and IL-8 levels were significantly lower in the LPR group compared with placebo (Figure S1 in S2 File).

Summarization of secondary outcomes was demonstrated in Table S4 in S2 File.

## Discussion

This randomized controlled trial provides robust clinical evidence that supplementation with a phytochemical-rich lime-based preparation (LPR) significantly reduces the recurrence of CaOx urolithiasis. Over a 24-month period, LPR intake was associated with a 76% relative risk reduction in stone recurrence compared with placebo. In addition, urinary biomarkers of renal injury demonstrated consistent improvement: protein excretion and IL-8 levels were significantly reduced in the LPR group, whereas no changes were observed in the placebo group. Importantly, no hepatic or renal toxicity was detected, supporting the safety of LPR for long-term administration. These findings highlight LPR as a culturally acceptable, well-tolerated, and cost-effective strategy for secondary prevention of CaOx stones.

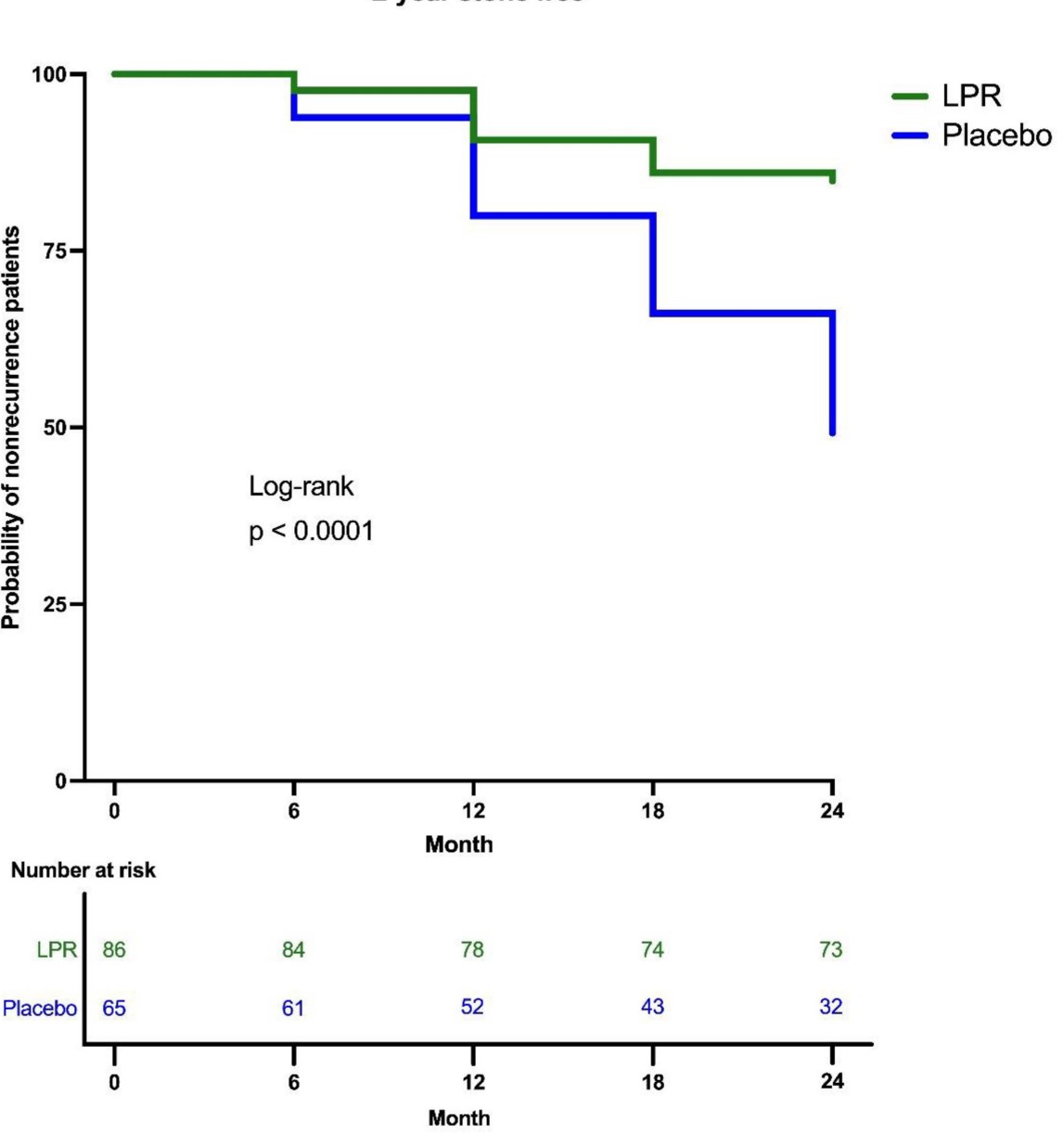

**Fig 2. Kaplan–Meier survival analysis demonstrated the time to stone recurrence in the LPR and placebo groups over a 24-month follow-up period.** Numbers at risk are displayed below the x-axis. The LPR group exhibited a significantly lower risk of recurrence compared with the placebo group, corresponding to an approximate 76% relative risk reduction (hazard ratio [HR] = 0.24; 95% CI: 0.13–0.44; p < 0.0001).

The observed clinical benefits are supported by mechanistic plausibility. Crystal-induced epithelial injury and micro-obstruction in stone disease promote local inflammation and cytokine release, including IL-8, thereby sustaining oxidative stress and facilitating crystal retention. Elevated urinary protein similarly reflects obstruction- or injury-related disruption of glomerular and tubular integrity. Reductions in IL-8 and proteinuria following LPR supplementation suggest attenuation of renal inflammation and restoration of barrier function, consistent with recovery of renal health [21]. Prior recommendations

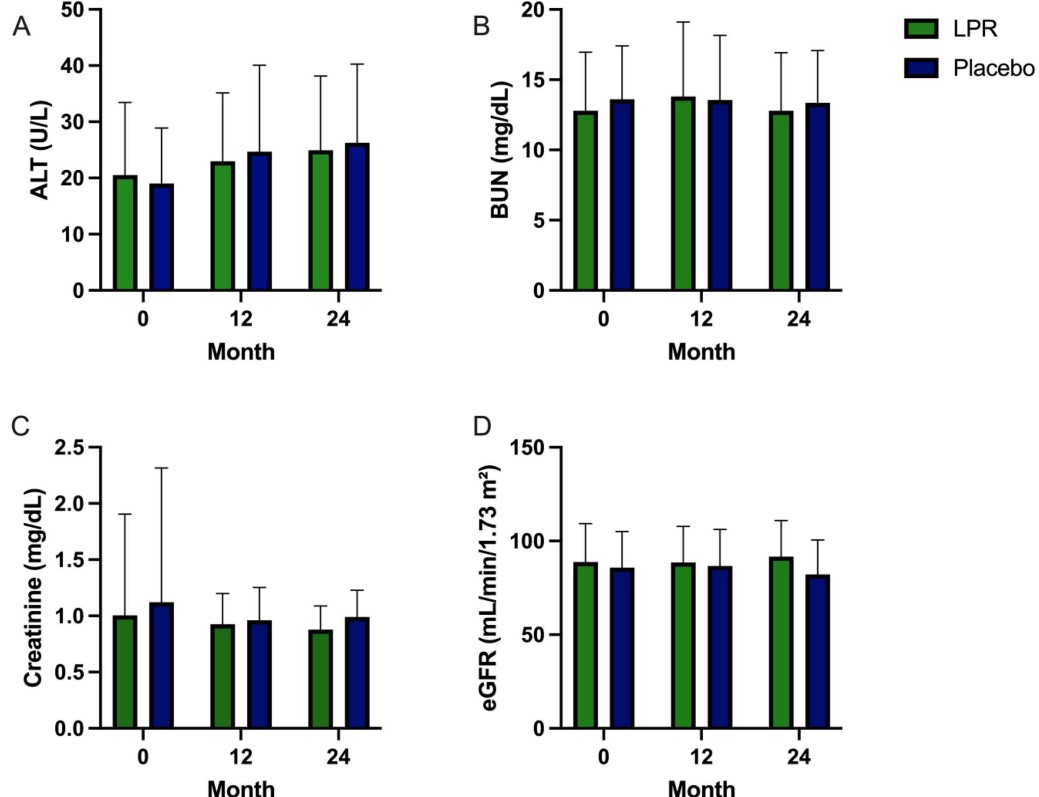

**Fig 3. Comparison of liver and kidney biomarkers between LPR and the placebo group at 0-, 12-, and 24-month follow-up.** (A) serum ALT, (B) serum BUN, (C) serum creatinine, and (D) estimated GFR.

https://doi.org/10.1371/journal.pone.0336892.g003

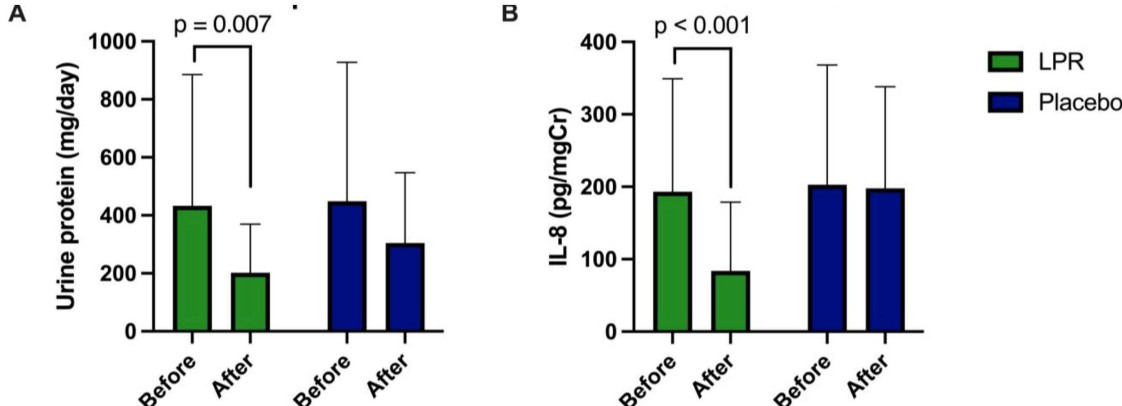

**Fig 4. Changes in 24-hour urinary variables between 0 and 24-month follow-up in LPR and placebo groups. (A)** Protein excretion was reduced within LPR group, but not placebo group. **(B)** IL-8 level was reduced within LPR group, but not placebo group. A pair t-test was used to evaluate, and the error bars represent standard deviations.

from the Experts in Stone Disease (ESD) Conference also emphasize the utility of proteinuria monitoring in high-risk stone formers, underscoring the clinical relevance of these markers [22].

Renal tissue injury is closely linked to calcium oxalate stone formation through a self-reinforcing cycle. Exposure to oxalate and calcium oxalate crystals induces reactive oxygen species, localized inflammation, and tubular injury [15]. In addition, injury to the renal tubular or papillary epithelium exposes underlying matrix components that provide adhesive sites for calcium oxalate crystal attachment. Once crystals adhere, they trigger oxidative stress and local inflammation, particularly through cytokines such as IL-6 and IL-8, which amplify tubular damage. Interstitial calcium phosphate deposits (Randall's plaques) exposed to urine can also act as a nidus for calcium oxalate overgrowth as well as activation of NLRP3 inflammasome, leading to maturation and release of IL-1β and IL-18 pro-inflammatory cytokines and cell death [16]. As calcium oxalate stones enlarge or pass, they cause further epithelial trauma and obstruction, perpetuating injury and increasing the risk of recurrence. Thus, renal tissue injury not only facilitates calcium oxalate stone initiation but is also exacerbated by stone growth and passage.

The protective actions of LPR are likely attributable to its combined effects on urinary chemistry and antioxidant defense. Previous studies from our group demonstrated that LPR enhances urinary citrate excretion and increases urine pH, mitigating lithogenic risk factors [18,19,23]. Furthermore, the high flavonoid content of LPR provides complementary antioxidant and anti-inflammatory effects in *in vitro* and animal studies [24,25]. Hesperidin has been shown to reduce oxidative stress and prevent renal injury *in vivo*; hesperidin and diosmin inhibit CaOx crystal growth and adhesion while preventing stone formation in animal models [26–28]; and eriocitrin exerts renoprotective effects against oxidative damage in models of diabetes, sepsis, and ischemia–reperfusion injury [29]. In addition, high dietary intake of phytonutrient antioxidants, including vitamin C, has been associated with reduced stone risk [30]. Although these phytochemicals have not yet been proven effective in the treatment of stone disease in humans, they represent promising nutraceutical candidates for stone prevention. Further clinical studies are required to confirm their efficacy. Taken together, citrate-mediated urine alkalinization, in combination with antioxidant activity, offers a multifaceted mechanism for the prevention of recurrent stones.

It is important to note certain considerations. Because LPR exerts a urine-alkalinizing effect, similar to potassium magnesium citrate, it may aggravate calcium phosphate stone formation (e.g., brushite) and is therefore not recommended for patients with this stone type [31,32]. Overall, LPR demonstrated a favorable safety profile, with only mild gastrointestinal effects such as dental hypersensitivity, and may be especially suitable for patients who experience frequent gastric discomfort. Additionally, the estimated cost of LPR is comparable to, or potentially lower than, that of standard potassium citrate therapy.

In summary, this trial demonstrates that LPR supplementation effectively reduces stone recurrence, improves urinary markers of renal injury, and is safe for long-term use. Its mechanistic basis—combining increased citraturia, urine alkalinization, and phytochemical-mediated antioxidant effects—positions LPR as a promising adjunct in the secondary prevention of CaOx urolithiasis. These results support the potential incorporation of phytochemical-based therapies into clinical practice guidelines, although further validation in larger cohorts is warranted.

This study has several limitations. First, complete urinary biochemical profiles were not available for all participants because of challenges in sample storage and transportation; however, our previous studies have consistently shown that LPR supplementation modulates urinary metabolic risk factors, supporting the biological plausibility of our findings [18,19]. Second, a higher-than-expected dropout rate occurred at the first follow-up, but this was effectively mitigated by engaging local village health volunteers, and no additional dropouts occurred thereafter. Third, although participants were advised on diet and fluid intake, individual compliance and dietary oxalate burden were not systematically assessed. Nonetheless, a subset analysis of 24-hour duplicated meals at 12 months showed no significant differences between-groups in energy, macronutrients, or micronutrients, suggesting that diet was unlikely to confound the outcomes. Fourth, urinary IL-8 and protein excretion, while useful markers of renal inflammation and injury, require validation as predictive biomarkers of stone recurrence in larger longitudinal cohorts. Fifth, the predominance of female participants in this study, despite the higher prevalence of urolithiasis among males, represents a limitation. Because many male patients were unable to

participate due to frequent work-related relocations, this sex imbalance may limit the generalizability of our findings to the broader population. Finally, randomization was not stratified by study location, leading to some imbalance in participant numbers across regions. Although baseline characteristics were comparable, the modest sample size may have limited our ability to detect subtle geographic differences.

## Conclusion

This study provides strong clinical evidence that LPR supplementation significantly decreases the risk of recurrent calcium oxalate stones. Over 24 months, participants receiving LPR experienced a 76% relative risk reduction in recurrence, with an absolute risk reduction of 35.7%, compared with placebo. These benefits were accompanied by improvements in urinary biomarkers of renal injury, including reductions in protein excretion and IL-8 levels. LPR was well tolerated over the 2-year trial period, with no hepatic or renal toxicity observed; however, long-term tolerability beyond this timeframe remains to be established. In accordance with current guideline recommendations, caution should be exercised in patients with advanced chronic kidney disease, particularly due to the potassium content of the supplement.

The clinical benefits observed in this study likely reflect attenuation of renal inflammation, while previous studies from our group have demonstrated that LPR also increases urinary citrate excretion and urine pH, supporting its mechanistic plausibility. While the flavonoid content of LPR may contribute to these protective effects, its role remains hypothesis-generating and requires further validation in human studies.

Taken together, our findings support LPR as a safe, culturally acceptable, and cost-effective intervention for secondary prevention of calcium oxalate urolithiasis, with potential for integration into clinical management strategies pending confirmation in larger, longitudinal trials.

## Supporting information

**S1 File. This file contains translated research protocol for the study, including study population, inclusion and exclusion criteria, discontinuation criteria, and intervention.**
(PDF)

**S2 File.** This file contains additional supporting materials for the study, including Table S1 (metal concentration in lime product), Table S2 (baseline demographic data), Table S3 (nutritional value of 24-hour duplicated meals), Table S4 (summarization of secondary outcomes and interpretation), and Figure S (24-hour urinary parameters between groups). These materials support the findings presented in the main text.
(DOCX)

## Acknowledgments

We thank Mr.Chainan Suksamit from Banpong Novitat Co.Ltd, and Mr.Wuthipong Tanakom from Oui Heng International Healthcare Co,Ltd. For assistance in the LPR and placebo manufacturing.

In the development of this manuscript, the author employed Writefull for grammatical corrections, and BioRender.com to create a figure. Following the utilization of this tool, the author meticulously reviewed and adjusted the content where necessary, assuming complete accountability for the final submission.

## Author contributions

**Conceptualization:** Thasinas Dissayabutra, Piyaratana Tosukhowong.

**Data curation:** Thasinas Dissayabutra, Piyaratana Tosukhowong.

**Formal analysis:** Thasinas Dissayabutra, Weerapat Anegkamol, Piyaratana Tosukhowong.

**Funding acquisition:** Piyaratana Tosukhowong.

**Investigation:** Wattanachai Ungjaroenwathana, Tasanee Klinhom, Thosaphol Sasivongsbhakdi, Pisitpol Siriwattana, Anuthep Burami, Ukrit Wayakkanont.

**Methodology:** Supoj Ratchanon, Pisit Prapunwattana, Piyaratana Tosukhowong.

**Project administration:** Supoj Ratchanon.

**Resources:** Thasinas Dissayabutra, Piyaratana Tosukhowong.

**Supervision:** Pisit Prapunwattana, Piyaratana Tosukhowong.

**Validation:** Thasinas Dissayabutra, Weerapat Anegkamol, Piyaratana Tosukhowong.

**Visualization:** Thasinas Dissayabutra, Weerapat Anegkamol.

**Writing – original draft:** Weerapat Anegkamol, Piyaratana Tosukhowong.

**Writing – review & editing:** Thasinas Dissayabutra.

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
