## [Decision Letter · Decision Letter 0]

23 Sep 2025

Dear Dr. Dissayabutra,

Thank you for submitting your manuscript to PLOS ONE. After careful consideration, we feel that it has merit but does not fully meet PLOS ONE’s publication criteria as it currently stands. Therefore, we invite you to submit a revised version of the manuscript that addresses the points raised during the review process.

Thank you for submitting the following manuscript to PLOS ONE.

Please revise the manuscript according to the reviewers' comments and upload the revised file.

We look forward to receiving your revised manuscript.

Kind regards,

Yung-Hsiang Chen, Ph.D.

Academic Editor

PLOS ONE

Journal Requirements:

2. We note that you have selected “Clinical Trial” as your article type. PLOS ONE requires that all clinical trials are registered in an appropriate registry (the WHO list of approved registries is at https://www.who.int/clinical-trials-registry-platform/network/primary-registries and more information on trial registration is at http://www.icmje.org/about-icmje/faqs/clinical-trials-registration/ ).

Please state the name of the registry and the registration number (e.g. ISRCTN or ClinicalTrials.gov ) in the submission data and on the title page of your manuscript.

a) Please provide the complete date range for participant recruitment and follow-up in the methods section of your manuscript.

b) If you have not yet registered your trial in an appropriate registry, we now require you to do so and will need confirmation of the trial registry number before we can pass your paper to the next stage of review. Please include in the Methods section of your paper your reasons for not registering this study before enrolment of participants started. Please confirm that all related trials are registered by stating: “The authors confirm that all ongoing and related trials for this drug/intervention are registered”.

Please see http://journals.plos.org/plosone/s/submission-guidelines#loc-clinical-trials for our policies on clinical trials.

[National Research Council of Thailand (Grant number 2556-29)]. 

4. Thank you for stating the following in your manuscript:

[This study was funded by National Research Council of Thailand (Grant number 2556-29). The funders had no role in the design of the study, collection, analysis or interpretation of data, writing of the manuscript, or the decision to publish the results.]

[National Research Council of Thailand (Grant number 2556-29)]

5. We note that you have indicated that there are restrictions to data sharing for this study. For studies involving human research participant data or other sensitive data, we encourage authors to share de-identified or anonymized data. However, when data cannot be publicly shared for ethical reasons, we allow authors to make their data sets available upon request. For information on unacceptable data access restrictions, please see http://journals.plos.org/plosone/s/data-availability#loc-unacceptable-data-access-restrictions.

6. Please include captions for your Supporting Information files at the end of your manuscript, and update any in-text citations to match accordingly. Please see our Supporting Information guidelines for more information: http://journals.plos.org/plosone/s/supporting-information .

Additional Editor Comments:

Thank you for submitting the following manuscript to PLOS ONE.

Please revise the manuscript according to the reviewers' comments and upload the revised file.

Reviewers' comments:

Reviewer's Responses to Questions

**Comments to the Author**

1. Is the manuscript technically sound, and do the data support the conclusions?

Reviewer #1: Yes

Reviewer #2: Partly

Reviewer #3: Partly

2. Has the statistical analysis been performed appropriately and rigorously?

Reviewer #1: No

Reviewer #2: Yes

Reviewer #3: I Don't Know

3. Have the authors made all data underlying the findings in their manuscript fully available?

Reviewer #1: No

Reviewer #2: Yes

Reviewer #3: Yes

4. Is the manuscript presented in an intelligible fashion and written in standard English?

Reviewer #1: Yes

Reviewer #2: Yes

Reviewer #3: Yes

Reviewer #1: I was unable to access or locate the recurrence data in the file provided via the Data Availability Statement (https://www.mediafire.com/file/su883wl3xfpqole/Raw+Data+LPR3.xlsx/file

). Please ensure that the raw data underlying the primary outcomes, particularly recurrence events and time-to-event data, are clearly available and accessible. Providing a well-organized dataset is essential for transparency and reproducibility.

Reviewer #2: As the statistical reviewer I will focus on methods and reporting.

Major

1) I could not see any power calculations in the main paper, please include at least a mention and that fully calculations are in the protocol.

2) it is supposed to be 1:1 but I see more people in the intervention. This does not seem to be linked to different drop out rates in the 2 groups. clarification is needed.

3) why wasn't the randomisation conducted within each centre for better balance?

4) A Cox model is mentioned in the methods section, with no covariates listed for adjustment - then KM curves are presented (which are fine if there is perfect balance across the covariates of interest - in this case there is no perfect balance on location despite this being listed as NS, since p-values are a function of sample size and in this case the sample size is low). So to summarise, A Cox model with location as a covariate would probably run and adjust for potential differences in diets that are location driven. if randomisation was done within stratum, this would not be an issue. If there is geographic variation in stone risk factors in Thailand, failing to adjust for location may introduce bias.

5) the analyses do not account for location. ideally this would be done using a random effect (shared frailty) in a Cox regression model, if the numbers allow.

6) The use of randomizer.org is mentioned, but more details are needed e.g. on allocation concealment and sequence generation etc

7) There are many secondary outcomes and the findings from these analyses should be explored with caution, considering the small sample size and the fact no corrections for multiple testing were applied. any findings need to be clearly be stated as exploratory and this needs to be discussed as a limitation in the relevant section.

8) The paper states that 151 out of 173 participants completed the study, but it’s unclear how dropouts were handled in the analysis. Clarify ITT was used, an important point since dropout rates were non-negligible.

Minor

1) report exact p-values to 3 decimals, don't just state NS

2) Some figures lack confidence intervals or number-at-risk tables, which are standard for survival analysis reporting.

3) The checklist marks several items as NA which contradicts the manuscript content.

4) The manuscript states no missing data for primary outcomes, but it’s unclear how missing data for secondary outcomes were handled.

5) If there is a taste difference between LPR and placebo, more detail on how blinding was assessed or maintained would be helpful.

6) A table summarising all secondary outcomes with effect sizes and confidence intervals would help readers interpret the findings more cautiously.

Reviewer #3: dear authors, the topic of your clinical trial is of great interest to the reader, and the manuscript overallis well written and concise. However I have certain objections and remarks related to the study methodology mainly. Please see below

1. why were patients with more than one stone excluded from the study?

were all patients with stones asymptomatic?

2.were the stones analyzed after lithotripsy, how do we know that they were all mainly of calcium oxalate stones? HU could help as proxy in that matter. Were HU measured for each stone and what was the threshold for considering a stone of calcium oxalate?

3.non contrast CT is the imaging of choice for detection of stones before and after lithotripsy

US and KUB are far less reliable especially for smaller stones, please explain why CT was not used for all patients consistently especially given that stone free rates after surgery cannot be reliably evaluated with US

4. how many patients had lithotripsy before study enrollement?

5.what is the amount of citrates delivered by LPR? compared to the suggested minimmum daily dietary intake? this information is missing

6.why ALT in particular was measured?

7. table 1: how is this percentage of calcium oxalate stones estimated since no relevant studies were done (stone analysis or use of HU?

8. discussion: discuss the clinical implications of elevated urine protein and IL8

9. discussion:what is the correlation between renal tissue injury and calcium stone formation?

**Do you want your identity to be public for this peer review?** For information about this choice, including consent withdrawal, please see our Privacy Policy

Reviewer #1: **Yes: ** Saba Jalali

Reviewer #2: No

Reviewer #3: **Yes: ** Petros Sountoulides

---

## [Author Response · Author response to Decision Letter 1]

1 Oct 2025

Response to reviewers’ comments

Comment Correction

Reviewer 1

Introduction

References: Some cited references are dated.

Lines 52–54: References 1–3 (incidence of urolithiasis) should be updated with more recent epidemiological data.

Line 56: References 4–6 supporting urinary metabolic abnormalities are dated and population-limited. Specifically, reference #6 appears less relevant, as it focuses on urinary citrate excretion and therapeutic interventions rather than directly linking urinary metabolic abnormalities to stone risk. In addition, the cited studies are relatively dated and limited to Thai populations, which may restrict their generalizability. Replace with more recent, high-quality reviews.

Line 59: The current references (6, 7) cited for the statement that “potassium citrate therapy is the standard pharmacological intervention to prevent recurrence” are limited, particularly reference #7, which is based on a small mechanistic study.I would also caution against relying exclusively on references (6, 7), as both are authored by the same research team and may introduce unnecessary self-citation bias. Given that the claim relates to standard pharmacological intervention, it would be more appropriate to cite authoritative clinical practice guidelines or comprehensive consensus statements. Like: European Association of Urology (EAU) Guidelines on Urolithiasis which recommend potassium citrate as first-line pharmacological therapy for recurrent calcium stone formers with hypocitraturia. or (AUA) Guideline on Medical Management of Kidney Stones

Line 60: References 8 and 9 (1985 and 2005) are outdated and do not reflect current pharmacological practice. Since then, potassium citrate formulations have evolved—for example, coated and extended-release preparations designed to reduce gastrointestinal side effects. Therefore, it would be important to replace or supplement these older citations with more recent, guideline-based and systematic review evidence. Stronger references should be provided to support adverse effects and adherence: recent studies highlight that, despite improved formulations, gastrointestinal intolerance and poor adherence remain major limitations, compounded by the high cost of long-term therapy, particularly in low-resource settings.

Lines 63, 60: Ensure periods are consistently placed outside reference parentheses.

Line 68: The claim that renal inflammation contributes to stone pathogenesis requires citation of recent studies.

Line 72: Please provide the citrate content of the lime-based regimen here.

Line 76: Expand on how IL-8 and urinary protein can serve as biomarkers of treatment response.

- We updated the reference to newer ones (reference no.1 to 3) as well as the corresponding statements in yellow highlights

- We updated the references to newer ones (Ref no 4-6) as your suggestion

-We updated the reference to the newer ones (Ref no.7-8) from EAU guideline and Nature review.

- We updated the references no. 9-10, however, even these manuscripts commented about gastrointestinal adverse effect of potassium citrate, but the trial performed in cystinuria and pediatric renal tubular acidosis, no urolithiasis. The latter is prolonged release of potassium citrate and potassium bicarbonate. Full stop is also corrected.

- We corrected the periods, and italic of C. aurantifolia

- We added references related to renal inflammation and oxidation-induced calcium oxalate stone formation (Ref no. 14-15).

- We added “Each sachet of LPR contained 7.75 g of supplement, providing a citrate dose comparable to standard alkali therapy but with a reduced potassium content.” in Methods section, LINE 104-109

- For proper arrangement, we added “The observed clinical benefits are supported by mechanistic plausibility. Crystal-induced epithelial injury and micro-obstruction in stone disease promote local inflammation and cytokine release, including IL-8, thereby sustaining oxidative stress and facilitating crystal retention. Elevated urinary protein similarly reflects obstruction- or injury-related disruption of glomerular and tubular integrity. Reductions in IL-8 and proteinuria following LPR supplementation suggest attenuation of renal inflammation and restoration of barrier function, consistent with recovery of renal health (20). Prior recommendations from the Experts in Stone Disease (ESD) Conference also emphasize the utility of proteinuria monitoring in high-risk stone formers, underscoring the clinical relevance of these markers (21).” in Discussion part, LINE 233-240

Methods

Line 85: Age eligibility is inconsistent: the protocol exclusion criteria mention ≤75 years, while the text states ≤70 years. Please clarify.

Line 89: The eligibility criterion in the Supplementary Material (residual stones ≤4 mm at 1 month post-surgery) is clearer than in the main text. Harmonize these.

Line 91: Sample size assumptions in supplementary material: In the sample size calculation section, you state that the historical recurrence rate is ~40% over 2.5 years and that the study anticipates a 40% reduction in recurrence risk (HR = 0.6), with calculations based on methods by Hsieh and Lavori, Schoenfeld, and Chow et al. However, no supporting references are provided for either the historical recurrence rate assumption or the anticipated effect size.

Line 94: Please include in the main text (not only in Supplementary Material) that each LPR sachet contained 63 mEq citrate and 21 mEq potassium. Compare this dosage with standard potassium citrate prescriptions (30–60 mEq/day). Note that this dose is at the upper end of recommended citrate therapy, but with less potassium. Providing this comparison would help readers better interpret the clinical relevance of the intervention relative to established therapies.

Placebo composition: Please report the exact lactose content per placebo sachet and discuss whether the amount falls below the usual tolerance threshold (≈12 g). Also clarify whether screening for lactose intolerance was considered, given its relatively high prevalence in Asian populations.

Line 105: Dose of 7.75 g LPR — please provide rationale (e.g., based on prior data, pharmacokinetics, or pilot testing). State explicitly how much bioavailable citrate this dose provides in mEq/day.

Line 115: The 24-hour urine protocol is unclear. Was urine collected only once at study end and baseline? Were duplicate collections performed to improve accuracy? Why was urinary citrate and other stone-related metabolites not reported?

Line 122: The safety monitoring process is clearly described; however, the handling of clinical events requiring intervention during follow-up is unclear. Please clarify: Management of recurrence requiring intervention: If, during the 2-year follow-up, a participant developed a stone of sufficient size or symptoms to warrant surgical removal or another intervention, how was this managed within the trial? Were such patients excluded from further follow-up analyses? Or were they retained and analyzed according to the intention-to-treat principle?

- We corrected the text in LINE 85 to “18-75 years”, following the protocol. Thank you for your suggestion.

- We corrected the statement into “had residual stones ≤4 mm at 1-month post-surgery evaluated by KUB x-ray or ultrasound.” LINE 88-89

- We appreciate your comment, but we could not find the information regarding your comment. For sample calculation we used “Medical management to prevent recurrent nephrolithiasis in adults: a systematic review for an American College of Physicians Clinical Guideline” Ann Intern Med. 2013 Apr 2;158(7):535-43. As mentioned in Supplementary. We added “A priori sample size and power calculations were conducted during the study planning stage. The calculation assumed a two-year stone recurrence rate of 40% in the placebo group and 15% in the LPR group, corresponding to a relative risk reduction of 0.25; 95%CI 0.14 – 0.44 (20). With a two-sided α of 0.05 and 80% power, the required sample size was 145 participants (72 per group). To account for an anticipated dropout rate of 10%, the target enrollment was increased to 160 participants. Full details of the power calculation are provided in the Supplementary.” In Study Design in Methods section, LINE 91-96.

- We were advised by our University’s Intellectual Property (IP) unit not to disclose the exact dosage of the main ingredient, particularly citrate, prior to patent approval, as this could potentially interfere with the patent granting process. For this reason, we have intentionally avoided emphasizing the precise amounts of citrate and potassium in the manuscript. As a compromise, we have revised the statement as follows: “Each sachet of LPR contained 7.75 g of supplement, providing a citrate dose comparable to standard alkali therapy but with a reduced potassium content. In addition, each sachet delivered 153 mg of flavonoids, including hesperidin, diosmin, and eriocitrin. Relative to conventional potassium citrate prescriptions (30–60 mEq/day), the citrate content in a single sachet falls at the upper end of the recommended therapeutic range, whereas the potassium load remains comparatively lower.” LINE 104-109. We wish the reviewers understand our restriction.

- The placebo had been changed to maltodextrin, as the consideration of high prevalence of lactose intolerance in elder participants. The protocol had been amended, and the detail of placebo was described in LINE 119

- We corrected the statement as mentioned above (LINE 104-109). The dose of 7.75 g LPR was determined based on the recommended daily intake of standard citrate treatment. As noted by the reviewer, we deliberately selected the upper limit of standard citrate therapy as the target dosage.

- It was our fault. 24-hour urine was collected twice. Because of this, we corrected the statement into “The 24-hour urine samples were collected at the 0- and 24-month”. LINE 133-134. Also, for consistency, we corrected the previous statement about blood collection into “Blood samples were collected at 0-, 12- and 24-month” LINE 130

- We added “Participants who developed urolithiasis with a stone size greater than 4 mm were withdrawn from the study and referred to a urologist for appropriate management.” LINE 141-142

Results (and Tables/Figures)

Line 147: The allocation of 98 participants to the intervention arm and 75 to the placebo arm suggests an unequal randomization ratio. Could the authors clarify How was the sample size determined, and did this imbalance arise by chance due to the randomization process? Loss of participants after randomization, allocation errors, or differential withdrawal before intervention).

Line 149: The phrase “number of stone and recurrence rate” is confusing — “stonr number” already mentioned in the sentence and “recurrence rate” at baseline is ambiguous since recurrence is typically a follow-up outcome.

Baseline data (Table 1):

-Consider adding stone size as a baseline variable.

-Note that males were underrepresented compared to expected prevalence. This is noteworthy, as epidemiological evidence consistently shows that the prevalence and incidence of urolithiasis are higher in men than in women. Could it indicate selection bias in enrollment? Could the predominance of female participants influence the generalizability of the findings, given known sex-related differences in stone risk and recurrence?

Type of stone: In the Methods section, there is no description of how stone types were determined.

Line 152: I recommend comparing the baseline characteristics of participants who were lost to follow-up with those who completed the study. This would help assess whether attrition introduced potential bias.

Line 166: The section on Detection of LPR contamination appears redundant, as similar information has already been provided under LPR manufacturing, quality control and supplementation. To improve conciseness, you could consolidate these details into a single section. For example, heavy metal and pathogen testing results could be summarized once under quality control, with Table S1 referenced there if needed, rather than repeating them in multiple places.

Line 173: The phrase “stone recurrent rate” is not standard; the appropriate term would be “stone recurrence rate”. I recommend revising the wording accordingly.

The manuscript states that stone recurrence was monitored by KUB radiography (6 and 18 months) and CT (12 and 24 months). However, this is more a description of the imaging schedule rather than a clear definition of recurrence. For clarity, please specify: Was recurrence defined as the appearance of a new stone, the growth of a pre-existing stone, or the need for clinical intervention? Additionally, please ensure that “recurrence rate” is defined consistently in the manuscript. Clear and consistent terminology will improve readability and scientific precision.

Line 198: A short comment in the Discussion on the clinical relevance of decreased urinary protein and IL-8 levels in relation to stone recurrence would strengthen interpretation.

Line 202: Please clarify whether the reported changes in urinary protein and IL-8 are (a) within-group changes from baseline or (b) between-group differences versus placebo. For efficacy inference in an RCT, the primary analysis should compare LPR vs placebo, adjusting for baseline values. report adjusted between-group differences (or ratios if log-scaled) with 95% CIs and P-values; avoid relying on within-group significance alone.

Line 203: A paired t-test is appropriate for within-group baseline vs. follow-up comparisons, but not for comparing intervention and placebo groups in an RCT. For efficacy inference, the main analysis should be a between-group comparison, ideally adjusting for baseline values

- We must acknowledge that we did not fully recognize the skewed distribution of participants between the LPR and placebo groups during the study. To minimize potential bias, we engaged a third party to generate the randomization sequence using the Research Randomizer website. We were not involved in this process until the study was completed, at which point we observed the imbalance. Our best hypothesis is that the randomization sequence, by chance, led to a greater allocation to the LPR group. Alternatively, it is possible that the third party applied an unequal randomization ratio without notifying.

- We removed “, number of stone and recurrence rate” as your suggestion

- Data on stone size could not be provided due to incomplete records.

- This finding was also of interest to us. Upon further investigation through participant interviews, we found that most male participants resided in suburban areas and were required to migrate for agricultural work for approximately four months each year and subsequently worked as laborers in the city for the remainder of the year. As a result, men were less able to commit to a two-year research study. In contrast, women who remained working on the farm were more available and willing to participate. Consequently, the number of female participants exceeded that of male participants.

- We added “The stone composition was analyzed by Fourier-transform infrared (FTIR) spectroscopy.” In Method section LINE 126

- We added “Table S2 Baseline demographic and clinical characteristics of participants between complete participants and drop-out” in supplementary

- According to your suggestion, we removed this section in Result, and rewrite the Method part as “The product was tested and certified by the Thai National Science and Technology Development Agency (NSTDA), which confirmed that heavy metals were below detection limits, which were within the acceptable limits for Daily Consumption of heavy metals - United States Pharmacopoeia (ALDC-HM USP) reference and no harmful microorganisms were present, ensuring the product's safety (Table S1).” LINE 113-117

- We corrected the “Stone recurrence rate” as your suggestion LINE 186 and recheck the rest of the manuscript.

- We added “The recurrence was diagnosed when the preexist

---

## [Decision Letter · Decision Letter 1]

20 Oct 2025

Dear Dr. Dissayabutra,

Thank you for submitting your manuscript to PLOS ONE. After careful consideration, we feel that it has merit but does not fully meet PLOS ONE’s publication criteria as it currently stands. Therefore, we invite you to submit a revised version of the manuscript that addresses the points raised during the review process.

We look forward to receiving your revised manuscript.

Kind regards,

Yung-Hsiang Chen, Ph.D.

Academic Editor

PLOS ONE

Journal Requirements:

Additional Editor Comments:

Thank you for submitting the following manuscript to PLOS ONE.

Please revise the manuscript according to the reviewers' comments and upload the revised file.

Reviewers' comments:

Reviewer's Responses to Questions

**Comments to the Author**

Reviewer #1: All comments have been addressed

Reviewer #2: (No Response)

Reviewer #3: All comments have been addressed

2. Is the manuscript technically sound, and do the data support the conclusions?

Reviewer #1: Yes

Reviewer #2: (No Response)

Reviewer #3: Yes

3. Has the statistical analysis been performed appropriately and rigorously?

Reviewer #1: I Don't Know

Reviewer #2: (No Response)

Reviewer #3: Yes

4. Have the authors made all data underlying the findings in their manuscript fully available?

Reviewer #1: Yes

Reviewer #2: (No Response)

Reviewer #3: Yes

5. Is the manuscript presented in an intelligible fashion and written in standard English?

Reviewer #1: Yes

Reviewer #2: (No Response)

Reviewer #3: Yes

Reviewer #1: Thank you for providing detailed and thoughtful responses to the previous comments. All points have been adequately addressed. There are only a few minor suggestions that could further improve the manuscript and enhance its clarity and precision.

In the paragraph beginning with “It is important to note certain limitations”, it would be more appropriate to change the term “limitations” to “considerations”. The points discussed in that section (e.g., the urine-alkalinizing effect of LPR, safety profile, and cost comparison) are not actual limitations of the study but rather important clinical or practical considerations related to the use of LPR.

In line 280–281, the statement “our previous studies have consistently shown that LPR supplementation modulates urinary metabolic risk factors, supporting the biological plausibility of our findings” requires an appropriate citation to substantiate this claim.

The explanation provided for the gender imbalance — that male participants were less available due to seasonal migration for work — is informative and contextually relevant. However, this should also be acknowledged as a limitation, since the overrepresentation of female participants may affect the generalizability of the study’s findings to broader populations, particularly to males who may differ in metabolic or lifestyle characteristics influencing stone risk or treatment response.

Reviewer #2: I struggled to follow the responses. can the authors provide clear, point by point responses for each specific question and comment?

Reviewer #3: congratulations for providing such an extended revision and having addressed all the reviewers comments

**Do you want your identity to be public for this peer review?** For information about this choice, including consent withdrawal, please see our Privacy Policy

Reviewer #1: **Yes: ** Saba Jalali

Reviewer #2: No

Reviewer #3: No

---

## [Author Response · Author response to Decision Letter 2]

28 Oct 2025

I added the previous Response to Reviewer #1, for the reviewer#2 part, with reorganized, into the new Response. I wish the reviewer could view the response better than last time.

---

## [Editor Report · Decision Letter 2]

3 Nov 2025

Lime-Based Supplement Reduces Calcium Oxalate Stone Recurrence: A Multicenter Randomized Controlled Trial

PONE-D-25-39899R2

Dear Dr. Dissayabutra,

We’re pleased to inform you that your manuscript has been judged scientifically suitable for publication and will be formally accepted for publication once it meets all outstanding technical requirements.

Kind regards,

Yung-Hsiang Chen, Ph.D.

Academic Editor

PLOS ONE

Additional Editor Comments (optional):

Congratulations on the acceptance of your manuscript, and thank you for your interest in submitting your work to PLOS ONE.
---

## [Editor Report · Acceptance letter]

PONE-D-25-39899R2

PLOS ONE

Dear Dr. Dissayabutra,

I'm pleased to inform you that your manuscript has been deemed suitable for publication in PLOS ONE. Congratulations! Your manuscript is now being handed over to our production team.

Kind regards,

on behalf of

Dr. Yung-Hsiang Chen

Academic Editor

PLOS ONE